# Vertically Aligned Nanowires and Quantum Dots: Promises and Results in Light Energy Harvesting

**DOI:** 10.3390/ma16124297

**Published:** 2023-06-09

**Authors:** Giuseppe Prestopino, Andrea Orsini, Daniele Barettin, Giuseppe Arrabito, Bruno Pignataro, Pier Gianni Medaglia

**Affiliations:** 1Dipartimento di Ingegneria Industriale, Università degli Studi di Roma “Tor Vergata”, Via del Politecnico, 00133 Rome, Italy; giuseppe.prestopino@uniroma2.it (G.P.); medaglia@uniroma2.it (P.G.M.); 2Università degli Studi “Niccolò Cusano”, ATHENA European University, Via Don Carlo Gnocchi 3, 00166 Rome, Italy; daniele.barettin@unicusano.it; 3Dipartimento di Fisica e Chimica—Emilio Segrè, Università degli Studi di Palermo, Viale delle Scienze, Ed. 17, 90128 Palermo, Italy; giuseppedomenico.arrabito@unipa.it (G.A.); bruno.pignataro@unipa.it (B.P.)

**Keywords:** nanowires, nanotubes, nanostructures, quantum dots, surface functionalization, wet chemistry, solar cells

## Abstract

The synthesis of crystals with a high surface-to-volume ratio is essential for innovative, high-performance electronic devices and sensors. The easiest way to achieve this in integrated devices with electronic circuits is through the synthesis of high-aspect-ratio nanowires aligned vertically to the substrate surface. Such surface structuring is widely employed for the fabrication of photoanodes for solar cells, either combined with semiconducting quantum dots or metal halide perovskites. In this review, we focus on wet chemistry recipes for the growth of vertically aligned nanowires and technologies for their surface functionalization with quantum dots, highlighting the procedures that yield the best results in photoconversion efficiencies on rigid and flexible substrates. We also discuss the effectiveness of their implementation. Among the three main materials used for the fabrication of nanowire-quantum dot solar cells, ZnO is the most promising, particularly due to its piezo-phototronic effects. Techniques for functionalizing the surfaces of nanowires with quantum dots still need to be refined to be effective in covering the surface and practical to implement. The best results have been obtained from slow multi-step local drop casting. It is promising that good efficiencies have been achieved with both environmentally toxic lead-containing quantum dots and environmentally friendly zinc selenide.

## 1. Introduction

Quantum dots (QDs) are nanometer-sized particles with sizes that are small enough to be comparable to the Bohr radius of the atoms that comprise their bulk material; usually, this represents no more than two orders of magnitude difference from the Angstrom unit [1]. To be applied in optoelectronic devices, QDs must be composed of semiconductor materials that offer high emission stability, optical gain, and size-adjustable wavelength shifts [2]. For example, early experiments on AlGaAs QDs on GaAs and InGaAs QDs on InP in the 1980s [3] proved to increase the optical gain by more than an order of magnitude. Optoelectronic devices that can be improved through the use of QDs are numerous, and their uses extend to biology and medicine because organic dyes are limited in exhibiting these properties and, therefore, are not suitable for many imaging and biosensing applications, such as:Lasers.Semiconductor optical amplifiers.Modulators.Photodetectors.Quantum computing.Fluorescence spectroscopy.Biomedical imaging.Biosensors.

Nanowires (NWs) are widely used in innovative electronic devices, especially in the form of dense arrays of vertically aligned nanowires (VANWs) with a very high surface area to volume ratio, a feature that is essential for improving performance in optoelectronic [4] and photochemical devices [5]. The opportunity to combine quantum properties with nanoscale physics through the external functionalization of NWs with QDs may offer unprecedented photocurrent conversions in solid-state devices. For example, using the above combination, the responsivity of photodetectors has increased by more than a decade [6,7]. Furthermore, since Sandhu [8] published a short piece in *Nature Nanotechnology* about a three-fold increase in the power conversion efficiency (PCE) of solar cells made with ZnO VANWs functionalized with PbSe QDs [9], compared to a simple thin-film version of ZnO [10] (see Figure 1), significant efforts have been made in the fabrication of these types of nanowire–quantum dot hybrid solar cells (NWQDSCs). Still using ZnO-based VANWs coupled with PbS QDs, the PCE of the solar cells increased by 35% [11]. These dramatic improvements are the result of the combined effects of the aspect ratio of the ZnO NWs and the density of the QDs. In subsequent studies, even higher PCEs were obtained on devices based on the combination of ZnO NWs and PbS QDs, up to 6% and 8% under 1 sun illumination [12,13].

Surface engineering does not always provide better results in light energy harvesting, as increasing the carrier path can increase the recombination of photogenerated charges. In [14], the nanostructured absorbent layer of Sb${}_{2}$S${}_{3}$ on the titania nanotubes conglomerates into medium-sized clusters when thermally annealed, increasing the carrier path; therefore, the efficiency of the nanostructured solar cell drops to 1.3% from the 2% value of the planar one. The complete functionalization of the NW surface with QDs is one of the main objectives that technology must pursue [15,16]. The use of QDs for the fabrication of flexible photovoltaic cells has already attracted a lot of attention due to the efficiency of QDs in collecting light; this was recently reviewed in 2019 and 2022, in terms of both efficiency and stability [17,18]. The use of NWs in solar cells has also recently been revised, mainly NWs of ZnO [19,20], due to the continuous progress in the wet chemistry of ZnO, which allows their easy fabrication on flexible and transparent substrates [21]. In this article, we will review the main optoelectronic materials, such as ZnO, TiO${}_{2}$, SnO${}_{2}$, used for surface engineering in photovoltaic cells functionalized with QDs and featuring improved PCEs. The goal is to provide the reader with a technical foundation to inspire the best strategies in terms of QD selection and nanowire functionalization techniques. This would enable NWQDSCs to match the performance of other types of solar cells, such as low-cost dye-sensitized solar cells, with PCE reported at around 12% under standard emulated sunlight [22], and recently pushed beyond the 15% threshold [23].

## 2. Quantum Dot Materials and Synthesis

QDs can be self-produced in the laboratory or purchased from chemical companies, such as Crystal Plex, Sony, Nagase, LG Display, Samsung Electronics, Thermo Fisher Scientific, DuPont, and many others. QD research is not limited to the academic environment, but the growing demand for better displays on smart TVs and mobile phones has prompted the industrial research system to invest a lot of resources in QDs. In fact, the global quantum dot market is expected to grow at a compound annual growth rate greater than 25% [24]. QDs are already available in many different semiconducting materials that are useful for optoelectronic devices, e.g., II–VI semiconductor compounds, such as cadmium selenide (CdSe), cadmium sulfide (CdS), cadmium telluride (CdTe), lead selenide (PbSe), zinc indium phosphide/zinc sulfide (ZnInP/ZnS), zinc cadmium selenide/zinc sulfide (ZnCdSe/ZnS); III–V compounds, such as indium arsenide (InAs); electronic materials, such as silicon or silicon dioxide; or bidimensional material, such as graphene, molybdenum disulfide (MoS${}_{2}$) or very promising perovskite quantum dots (PQDs) still with some stability issues to be solved for large–scale use in commercial products. Among all of the used semiconducting materials, given the requirement of using environmentally friendly materials in large-scale industrial processes, there is a lot of interest in indium phosphide QDs [25]. Regarding innovatively advanced optoelectronic materials, it is worth mentioning biocompatible and highly stable diamond QDs. Their use in molecular bioimaging is already well proven [26]. Industrial research mainly focuses on light–emitting devices. For example, the Samsung Research Institute (SAIT) recently developed a blue–emitting ZnSe QLED with a 2–year lifespan at 100–nit luminance [27]. Some months ago, Zhijing Tech presented a prototype of a 75${}^{\prime \prime }$ inch television based on perovskite QD-colored pixels [28]. Thanks to these outstanding achievements, academic research has the opportunity to deploy low-cost industrial QDs for other important applications, such as photovoltaic energy. In the next two subsections, we will describe the two main techniques to synthesize QDs: the sol–gel route and organometallic deposition.

### 2.1. Sol–Gel Method

The cheapest and simplest method to synthesize QDs is based on sol–gel wet chemistry, where the chemical precursors react in a solution [29,30,31,32]. Colloidal solutions allow for multiple molecular reactions within the liquid volume and the kinetics can vary greatly. Given the size-dependent properties of QDs, it is necessary for the size distribution of the nanoparticles to be as narrow as possible (i.e., monodispersity). This requires the simultaneous inactivation of chemical reactions on the surface of the QDs within the solution. It is usually obtained by introducing capping molecules that, based on their physicochemical properties, confer the following attributes:Narrow size distribution.Solubility.Chemical inertness.Dielectric barrier.Surface trap filling.

The monodispersity requirement is especially true for light-emitting devices but is not as strictly necessary for light-harvesting applications, making the sol–gel method very suitable for fabricating QDs for NWQDSCs. In fact, in NWQDSCs, it is sufficient for the high energy level of the QD to be closer to the vacuum level than the conduction band of the electron transport layer (ETL), and that the reverse happens for the low energy level and the valence band of the hole transport layer (HTL). On the other hand, it is much more important to consider the effects of the capping molecule on the QD shell interface. The typical procedure involves dissolving a salt containing the desired elements that make up the QD, e.g., zinc acetate for ZnO QDs [33], and then slowly adding a basic solution (e.g., potassium hydroxide in ethanol) cooled to low temperatures (~5 ${}^{{}^{\circ}}$C). Subsequent surface functionalization can be performed by mixing the sol–gel solution of QDs with DI water mixed with other molecules. QDs should be separated from unreacted precursors by centrifugation, ethanol washing, and drying.

### 2.2. Organometallic Synthesis

Organometallic compounds have been extensively employed in metal–organic chemical vapor deposition (MOCVD) for the growth of single crystalline films with thicknesses within the range of a QD diameter of 1–10 nm, regardless of their nature, i.e., insulators, semiconductors, or metals. This versatility originates from the large variety of organometallic compounds in nature. Organometallic synthesis is based on the hot-injection method, which involves the reaction between nutrient solvent solutions at high temperatures using non-polar solvents, mainly trioctylphosphine oxide (TOPO). This technique, even if it is more complex and costly, is able to produce higher quality QDs [34,35,36,37], with superior physicochemical properties, such as:Long-term stability.Monodispersity.High-quantum efficiency.Long luminescence lifetime.Electron paramagnetic resonance inactivity.

Organometallic techniques may also be used for the preparation of the capping layer of QDs. For example, using hydrophobic molecules, such as 2-(2-methoxyethoxy)acetate, may make ZnO QDs resistant to both chemical and biological environments [38].

### 2.3. Physical Deposition Systems

The initial research on devices based both on NWs and QDs assumed their integration during the NW growth. This technology has been well studied and documented [39] as it is essentially based on the synthesis of heterostructures through physical deposition techniques, such as molecular beam epitaxy [40], pulsed laser deposition [41], chemical vapor deposition [42], droplet epitaxy [43], and even multiple combinations of the aforementioned techniques [44]. In particular, physical epitaxial techniques allow for the strong control of material growth and, therefore, of QD lattice embedded into the nanowire body or grown onto its external facets with inherent mechanical strain, see Figure 2. As an example, InAs QDs grown on GaAs NW facets have demonstrated the availability of both single and double excitons in low-temperature photoluminescence spectroscopy [45,46]. This behavior is well modeled, considering the electromechanical field effects for tight-binding atomistic calculations [47]. On the other hand, such techniques usually require high temperatures and costly equipment and are more suitable for the production of highly performant photonic devices rather than cost-competitive solar cells [48].

### 2.4. Quantum Dots Properties

QDs are very small nanoparticles with diameters of less than 20 nm; they may undergo electron quantum confinement, which leads to a change in the band gap energy of the bulk semiconductor material since the state distance in the k–space *(Δk)* proportional to the size of the QD *(L${}_{qd}$)*:(1)$$\Delta k=\frac{2\pi }{L_{qd}}.\;\;E_{qc}=\frac{h^{2}}{2m^{*}L_{qd}^{2}}$$
where $m^{*}$ is the exciton effective mass:(2)$$\frac{1}{m^{*}}=\frac{1}{m_{e}^{*}}+\frac{1}{m_{h}^{*}}.$$

For example, the widely studied silicon QDs are able to increase the energy level gap by 0.3 eV when halving their diameter from 8 to 4 nm [50]. A sketch of the gap change due to quantum confinement is reported in Figure 3.

In QDs, it is possible to observe multiple exciton generation when incoming photons are able to create more than a single electron–hole pair. QDs possess a built-in dipole due to the spatial separation of positive and negative charge carriers. Generally, the electrons concentrate in apex regions and the dipole points from the apex to the base. It is possible to prove it through mathematical modeling of the red shift of emitted photons under a dc bias or by measuring the dielectric dispersion, resulting in a dipole moment of about 10 eÅ for very small QDs (r < 2.5 nm) [51] or tens of eÅ for larger sizes [52].

## 3. Vertically Aligned Nanowire Synthesis

To produce a nanostructured layer as a 3D scaffold for absorbing light through sensitization with QDs, researchers mainly employ the following three semiconductor oxides:

Titanium dioxide (TiO${}_{2}$).Zinc oxide (ZnO).Tin oxide (SnO${}_{2}$).

As can be seen from the physical properties summarized in Table 1, ZnO, which is the most used, as well as SnO${}_{2}$, have a higher mobility than TiO${}_{2}$; the latter is a low-cost and easy-to-process direct band gap semiconductor with a band structure and physical properties similar to ZnO. It is still receiving attention from solar cell researchers, as will be shown in the dedicated section; however, for the NWQDSC structures analyzed in this review, it is probably the least suitable material to choose.

In addition to mobility, the exciton lifetime is also important to consider in the nanostructured versions of these semiconductor materials. It is worth noting that in the literature, a decrease in the recombination lifetime has been reported for TiO${}_{2}$ nanotubes compared to TiO${}_{2}$ nanoparticles [57]; however, the inverse effect has been demonstrated for tin oxide nanowires [58], making them very strong competitors to ZnO, but with a more complex manufacturing process.

### 3.1. TiO${}_{2}$

This material, especially in the form of nanotubes, is widely used as an electron transport layer in dye-QD-sensitized solar cells; however, the crucial problem of electron mobility, as reported in Table 1, is partially compensated for by the very long recombination lifetime, which guarantees a good charge extraction at the metallic contacts [59]. The low charge mobility in standard TiO${}_{2}$-based dye-sensitized solar cells (DSSCs) is due to the charge transfer at grain interfaces [57], a problem that is avoided in vertically aligned nanowires or nanotubes. The latter form is preferred for nanostructured TiO${}_{2}$ electrodes, both due to the greater sensitizing surface available and the consolidated and simple electrochemical synthesis route [60,61].

The TiO${}_{2}$ nanotubes obtained through the anodization of titanium plates in electrolytes containing fluoride ions can be produced with precise control over their diameters (see Figure 4) and strong confinement between the different nanotubes, giving life to highly ordered arrays of vertically aligned nanostructures [62]. This procedure, in addition to being relatively simple, is a cheap and easily scalable method and it mainly requires:Application of a tunable DC bias voltage to the titanium sheet (i.e., 10–20 V).An electrolyte solution (i.e., 0.1–0.4 M HF).Equilibrium between surface oxidation and dissolution rates.Continuous growth into vertically aligned nanopores or nanotubes (rate 0.5 μm/h).Annealing in the air with slow heating and cooling rates (1–5 ${}^{{}^{\circ}}$C/min).

**Figure 4 materials-16-04297-f004:**
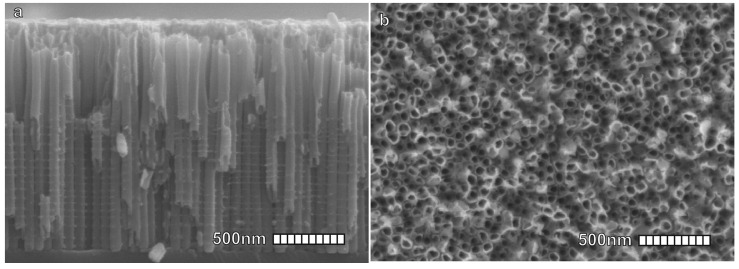
Scanning Electron Microscope (SEM) images of TiO${}_{2}$ nanotubes developed for DSSCs; (**a**) cross-Section. (**b**) Top view. Reproduced with permission from Ref. [57]. Copyright 2015, ACS pubs.

By changing the chemical bath from dilute hydrofluoric acid to aqueous solutions of buffered fluorine salts, and increasing the anodizing time and voltage, it is possible to obtain ordered nanotubes of TiO${}_{2}$ with aspect ratios up to 1000, traversing the entire thickness of the original titanium plate to achieve freestanding nanotube membranes. Subsequent treatment with titanium chloride may enable the complete formation of TiO${}_{2}$ nanorods, if desired [63]. TiO${}_{2}$ nanostructures can even be synthesized from other material templates, such as ZnO, whose growth will be introduced in the next subsection. The liquid-phase deposition involving ammonium hexafluorotitanate hydrolysis enables the fast deposition of TiO${}_{2}$ on the ZnO nanorod shell and slow etching of the ZnO nanorod core [64].

### 3.2. ZnO

ZnO is a wide band gap (3.37 eV) semiconductor and has outstanding potential for optoelectronics in the near ultraviolet (UVA) range. Furthermore, ZnO nanowires, which belong to the wurtzite crystal group and exhibit a large piezopotential when mechanically stressed due to the non-centrosymmetric lattice, have been shown to be effective at increasing the charge collection efficiency in flexible solar cells due to the induced strain [65,66]. The realization of vertically aligned ZnO nanowires on different conductive surfaces is extremely simple and efficient, thanks to the discovery of galvanically assisted nucleation [67,68] in a standard equimolar solution of zinc nitrate and triethylamine. This technique allows for a large-scale array of vertically aligned nanowires to be fabricated at a low cost, even if the overall length is limited to a few microns, well beyond the length used in solar cell applications, usually between 1 and 2 μm. However, there are many techniques used to increase the length of the grown ZnO nanowires, such as the addition of polyethylenimine (PEI) [69], the dynamic optimization of the temperature [70], or the use of other more expensive strategies, such as engineered thermoconvective growth in a solution [71], in which the nutrient solution is thermally stimulated by a local heater whose surface temperature is monitored, thanks to a small Pt100 temperature sensor inserted into a small hole drilled in the heater, and the synthesis in high-temperature ovens [72]. As will be explained in the section on solar cell performances, an excessive NW length can be detrimental because the charge collection efficiency decreases beyond a certain length, but the scattering length of the electrons, estimated by intensity-modulated photovoltage spectroscopy or the rise/fall times of the photocurrents induced by laser pulses, can be equal to 40–70 μm [59], indicating that the electron collection efficiency in NWQDSCs cells can be ensured with a thin film of long NWs (see Figure 5d). However, there is no research work in this direction and researchers have concentrated on the much more important problem of surface passivation [73].

### 3.3. SnO${}_{2}$

SnO${}_{2}$ is a wide band gap material (E${}_{g}$ = 3.6 eV) with very high electron mobility and, therefore, it is suitable for the fabrication of nanostructured photoanodes in solar cells. The array of SnO${}_{2}$ nanorods for solar cells can be implemented with high-temperature physical deposition systems, such as pulsed laser deposition [74] and droplet epitaxy [75,76], or with low-temperature wet chemistry recipes [77,78,79]. It is also noteworthy that by employing chemical vapor synthesis at very high temperatures (>1000 ${}^{{}^{\circ}}$C), it is possible to obtain well-aligned SnO${}_{2}$ nanotube arrays. In Figure 6, we reproduced SEM images of dye-sensitized solar cell photoanodes realized using this nanostructured material. These photoanodes exhibited a PCE of 2.53% [80].

### 3.4. Quantum Dots Sensitizing Techniques

The main technique used for dispersing QDs on a surface consists of mechanically pressurizing a viscous solution of QDs through a sliding blade, and regulating the thickness of the applied layer by regulating the three main parameters: distance, speed, and temperature. On the other hand, it is evident that this method is not suitable for nanostructured surfaces with high aspect ratio nanowires, which are too brittle against the applied lateral forces. In fact, many other techniques exploit the unique and adjustable viscous liquid–solid interactions to produce, after the evaporation of the solvent, heterogeneous layers of different materials on a substrate with controlled thickness. In addition to blade casting, the main techniques for functionalizing surfaces with QDs are:Bar coating.Zone casting.Spin coating.Drop coating.Spray coating.Dip coating.Langmuir–Blodgett.

These methodologies depend on a unidirectional viscoelastic interaction between the substrate surface and the colloidal solution, which in our case should contain dispersed QDs. Bar coating is the main technique used in industrial production of large–area devices with organic molecules as active surface layer [81] and it schematically shown in Figure 7 regarding.

Even if bar coating is suitable for controlling the molecular orientation, it is easy to implement and allows for multiple depositions, in the case of NWQDSCs, the solution’s shear forces must be adjusted so as not to damage the nanowire array; thus, the control of the meniscus of the solution adjacent to the substrate surface could be a very challenging step. Setting up a procedure for the QD sensitization of the VANW array by bar coating should be the final step for the production of low-cost NWQDSCs to commercialize. In the following subsections, we will focus on spray and dip coating techniques, mostly used in the academic research environment for the fabrication of solar cell prototypes.

#### 3.4.1. Spray Coating

The process of spray coating the QDs to sensitize the nanostructured photoanodes is even more complex. Indeed, this technique, when used to fabricate organic solar cells, has often produced poor energy conversion efficiencies (<5%) attributable to the difficulty in creating a good interconnection between the nanoparticles and the solar cell surface [82,83]. Regarding this technique, in 2015, Prof. Kramer’s group published a fundamental work [84]. The study reported a room-temperature spray coating process of PbS QDs, demonstrating the control of nearly a single layer. The method yielded solar cells with an impressive performance of 8.1%. The overall manufacturing process, however, is very slow, and requires between 65 and 85 spray sequences (see Figure 8), where each sequence consists of the following procedures (just under one minute) for a total spray time of over an hour:0.4 s QD monolayer spraying.3 s pause.1 s mercaptopropionic acid-spraying.4 s methanol spraying.40 s air blade drying.

**Figure 8 materials-16-04297-f008:**
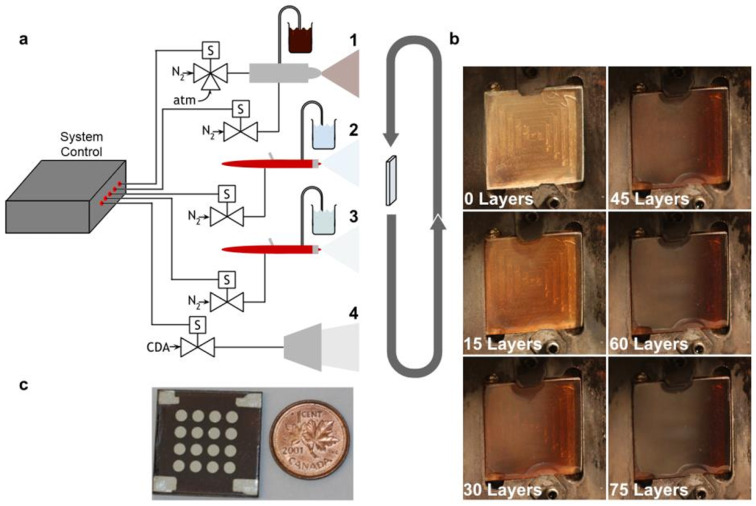
(**a**) Spray coating setup with three different nozzles for the three different spraying solutions: (1) fine mist spraying of colloidal QDs; (2) standard MPA–spraying; (3) standard methanol–spraying and (4) air drying. (**b**) Image of the QD sprayed coating according to the different number of sprayed layers. (**c**) Final device, ensemble of sixteen different sampling electrodes over its surface. A one-cent coin is shown for dimension reference. Copyright 2015, [84] Wiley.

#### 3.4.2. Dip Coating/Langmuir–Blodgett Deposition

Dip coating is one of the most effective thin film deposition processes. As the name suggests, it is based on the controlled immersion of the substrate in a tank containing the desired coating materials dispersed in a colloidal solution. As can be seen in Figure 9a, the substrate is immersed in the solution at a certain speed (usually rather slowly), then a rest time must be implemented, which allows the formation of a capillarity bond at the substrate/solution interface (meniscus). The substrate is withdrawn from the vessel at a specific speed, which must prevent evaporation at the upper edge from changing the profile of the meniscus. Dip coating is a very effective process for fabricating NWQDSCs, as the substrate is vertically immersed in the solution and there is no risk of compromising the brittle array of vertically aligned nanowires. It is also low-cost and industrially scalable (see Figure 9). The main parameters to control the deposition are viscosity and surface tension. Dip coating was the technology used for the fabrication of the first NWQDSC [9], which involved the integration of 2 nm PbSe QDs on ZnO VANWs.

With the appropriate hydrophobic ligands, it is possible to locate QDs at the water/air interface with a single monolayer thickness. In this way, dip coating evolves into the Langmuir–Blodgett deposition, which allows transferring very dense monolayers of molecules to the desired target thanks to the control of the lateral and surface pressures on the Langmuir monolayer during the emergence of the substrate. The Langmuir–Blodgett technique is particularly suited for precisely coating nanometric rough surfaces with monolayers of QDs, as demonstrated in several papers [85,86]. On the other hand, it requires the functionalization of QDs with complex ligands, which can influence their optoelectronic properties due to potential vibroelectronic energy exchanges [87], which influence exciton dynamics and charge transport [88]. Moreover, to prevent compromising charge tunneling, it is usually preferable to use ligands with the shortest possible atomic chains [86].

### 3.5. Electronic Transport Properties of Surface-Functionalized Nanowires

Particular attention must be paid to the electron transport properties at the interface between the array nanowire and adjacent working QDs. If we consider a generic nanowire at the interface with a dielectric medium, there is a space charge region (shell in Figure 10a), due to the electric field created by the bending of the conduction band away from the Fermi level. In the case of semiconductor oxides (for example, ZnO nanowires), this phenomenon is enhanced by the high rate of reaction of the defective surface with the oxygen molecules present in the atmosphere, which, once absorbed on the surface of the nanowire, tend to accept the free electrons to form anions and, therefore, increase the depletion region (shell in Figure 10b). The influence of the partial pressure of oxygen on the depletion regions of ZnO nanowires and nanobelts has been confirmed by several studies [89,90,91], which certify the increase of the conductivity under vacuum conditions due to the desorption of oxygen, leading to increased mobility by avoiding surface traps [92]. Chemical adsorption on ZnO nanowires does not always have the same effect as oxygen molecules, i.e., as acceptors that capture electrons. For instance, a strong interaction between ZnO nanostructures and ambient humidity has been reported due to the chemisorption of hydroxyl groups on their external surfaces, which behave as donor molecules and increase the availability of electrons in the conductivity band [90].

At standard atmospheric pressure, oxygen chemisorption at the rod surfaces captures free electrons (Figure 10b), increasing the space charge region and enhancing the distance between the Fermi level and the conduction band. This effect is detrimental to the efficiency of NWQDSCs efficiency since the nanowire layer usually acts as a photoanode and it needs to collect generated electrons. This is due to the work functions values of the used semiconductor materials. Indeed, as illustrated in the results reported in the next section of the review, the highest light harvesting efficiencies in NWQDSCs were obtained by creating a thin insulating layer on the nanowire surface prior to quantum dot functionalization and employing charge tunneling transport between the QDs and the NW body. Furthermore, it is worth noting that the promotion of electrons from the valence band to the conduction band directly within the bulk of the nanowire enables the tuning of the energy gap between the electron bands of QDs and NWs. In fact, when a light source creates electron–hole pairs, they can move by drift due to built-in fields or by diffusion. In a semiconductor, the majority carriers can drift at a greater length compared to minority carriers, but, in a thin nanowire body, they can reach the depleted shell in large numbers. As illustrated in Figure 10a, ZnO grown using standard wet chemistry recipes is an n-type semiconductor [93,94]. The photogenerated holes will then recombine with the adsorbed negative oxygen ions, which will be desorbed from the rod surface. This process reduces the energy barrier of a voltage, known as surface photovoltage (SPV), as depicted by the curly braces in Figure 10a; it also reduces the depletion volume and counteracts the drift of electrons caused by the built-in field. This behavior has been confirmed by photochemical studies on ZnO nanorods with functionalized surfaces [95], generating a lower SPV signal under the influence of UV illumination. Thus, after wet functionalization with QDs, as illustrated in previous subsections, by intense illumination at the appropriate wavelength, it would, in principle, be possible to change the surface potential by a desired amount.

## 4. Nanowires–Quantum Dot Solar Cells

To compare the performance of NWQDSCs in the literature, it is appropriate to consider the previous reviews that focused on similar nanostructured photoanodes [20,96] and quantum dot-sensitized solar cells (QDSSCs) [17,18], which, similar to dye-sensitized organic solar cells (DSSC), have a light-absorbing medium (dye or QD) embedded by two different materials, i.e., electron/hole transport materials or layers (ETM/HTM or ETL/HTL, respectively), dedicated to the transport of each excited charge (electron or hole). Such a device configuration would enable overcoming the transport limitations observed in QD thin films [97] assembled in standard quantum dot solar cells (QDSCs), where, to reach the metal contacts, both carriers drift along a thick layer of heterogeneous QDs, becoming trapped in the larger ones (lower band gap), resulting in a higher recombination rate. In the following Table 2, as a starting point for the analysis proposed in this work, we present a selection of the best articles mentioned in previous reviews.

It is worth noting that interpenetrating NWQDSCs showed the best performance when fabricated using ZnO NWs and PbS QDs (PCE = 9.6% @ row 3 of Table 2). Moreover, in [104], the performance further improved to 10.8% by utilizing ZnO NWs with optimized areal density and surface treatment. This result is not surprising considering that in this type of solar cell the light harvesting involves the QDs ensemble and the light collection increases with the volume it occupies. It is important to note that in standard QDSCs (without interpenetrating NWs) the performances are much better (PCE = 15.3% [108], see the last row of Table 2). In this work, Song et al. focused on optimizing heterojunction bindings with a double QD deposition approach on TiO${}_{2}$ nanoparticles, using capping ligand-assisted self-assembly and on-purpose metal oxyhydroxide adsorption sites. In this way, they were able to form a light-absorbing mesoporous structure (see Figure 11) that was capable of exceeding the 15% PCE threshold, which is the same record mentioned in the introduction achieved by standard DSSCs without QDs utilization. It is also important to note that the researchers employed Pb- and Cd-free “green” QDs: MgCl${}_{2}$ capped Zn–Cu–In–S–Se (ZCISSe) QDs.

To understand the development possibilities of NWQDSCs, it is enough to mention that the best improvements to NWQDSCs have been made by Dr. Tavakoli’s group at the Massachusetts Institute of Technology, using the combination of ZnO NWs–PbS QDs [105,106] (see Figure 11a). In 2019, they implemented a UV light down-shifting layer (CdSe/ZnS QDs) [105], and in 2020, they optimized the surface area and areal density of NWs [106]. However, these very brilliant techniques have never been combined together in a new solar cell. Other recent studies report the use of different types of nanowires [110] or quantum dots on NWs of ZnO [111], but with poor results (PCE < 1%); therefore, we concentrate our further comparative analysis on solar cells with nanowires embedded in perovskite layers or functionalized with perovskite QDs (PQDs) (see Table 3 and Table 4).

Moreover, when considering solar cells based on single-junction synthetic halide perovskites that utilize planar defect-engineered semiconducting charge transport layers, the overall PCE increased from 3.8% to 25.5% between 2009 and today [116]. In this context, the effect of interpenetrating ZnO NWs does not seem to be so positive if only the best overall efficiency results are sought. For example, in [117,118], using as ETL, respectively, a CuCl${}_{2}$/MDACl${}_{2}$ superficially modified tin oxide planar film, PCEs of 20.3% and 22.3% were obtained as the best efficiency values, even with excellent temporal stability and reversible behavior. Moreover, as can be seen in Table 3, the efficiency of perovskite solar cells using NWs (all ZnO–based) does not exceed the threshold of 20%. On the other hand, in all of the cited works, the positive effect of the nanostructured electrode compared to the planar one is well understood. For example, in [104], the planar version has a 25% lower short-circuit current than the nanostructured version. Indeed, wide bandgap nanowires with high electron and hole mobility, when used as photoanodes/cathodes, are expected to strongly favor the collection of charges excited by the photon-absorbing layer. Thus, we attribute the fact that the efficiency of NWQDSCs is not yet superior to that of solar cells with other architectures to the complexity of combining accurate models of the embedded QDs carrier transport mechanisms [119] with the complex light-trapping effects of the three-dimensional structure [120]. Even the ability to control the non-perfect uniformity of samples obtained through low-cost nanofabrication processes with wet–chemistry recipes can play a fundamental role [121].

**Table 4 materials-16-04297-t004:** Best papers on solar cells with the addition of perovskite quantum dots improving PCE and stability extrapolated from the review [122].

Fabrication Technique	Reference	J${}_{SC}$	V${}_{OC}$	PCE
Type of PQDs	Year	[mA/cm${}^{2}$]	[V]	[%]
Spin coating	[123]	23.4	1.15	18.3/21.5 *
$MAPbI_{3}$	[2019]			
Spin coating	[124]	23.86	1.11	18.4/19.5 *
$Cs_{0.57}FA_{0.43}PbBr_{3}$	[2019]			
Spin coating	[125]	23.4	1.14	19.5/21.1 *
$CsPbBr_{1.85}I_{1.15}$	[2019]			
Spin coating	[126]	23.8	1.07	18.2/20.06 *
$Cs_{0.05}{\left(MA_{0.17}FA_{0.83}\right)}_{0.95}PbBr_{3}$	[2020]			

* Increased Time-Stability to Humidity.

In Table 4, extrapolated from the review [122], we present the results obtained from the application of PQDs in solar cells. Again, the overall efficiency is not at the level of bulk perovskite-based solar cells [127]; however, this type of QD is able to increase the light-harvesting efficiency as a frequency down-shifting layer for c-Si solar cells [128], and above all, act as a stabilizing interlayer between the perovskite and the HTL material. Thus, it holds great promise as a powerful tool for the advancement of new generations of NWQDSCs. As the last point to underline, we highlight the potential of fabricating piezoelectric VANWs as photoanodes of flexible solar cells, as the role of the piezopotential at the ETL interface has been shown to effectively increase the PCE by more than 13% (see Figure 12). This result is higher than predictions of complex nanofabrication techniques for light trapping, which, at best, promise an increase in efficiency of no more than 11% [129].

## 5. Conclusions and Outlook

Solar cells based on the combination of semiconducting NWs and QDs have not yet received a dedicated review; they have only been mentioned in subsections of more general reviews on QDSCs, QDSSCs, or single-material ones (e.g., ZnO). Thus far, the best results have been obtained through the implementation of ZnO NWs, whose geometric characteristics (density, length, morphology, etc.) play important roles in the overall efficiency. Lead sulfide QDs have been shown to be the most effective light harvesters, with nanowire surface passivation as an essential manufacturing step to prevent current leakage at the interface between ZnO nanowires and PbS QDs. There are other promising but complex approaches to harvesting light energy, which are theoretically capable of exceeding the Shockley–Queisser limit, such as thermionic emitters with solar concentrating systems, but they are still at an embryonic level (PCE < 10% [130]). Research on NWQDSCs, which began 14 years ago with a PCE of 2%, has rapidly progressed to achieve a 5-fold increase, surpassing efficiencies greater than 10%, presenting the opportunity for NWQDSCs to be a part of multi-junction solar cells [131] as an additional design approach to surpass the Shockley–Queisser limit. The 10% goal, even if is not in line with the best results obtained on solution-processed planar solar cells, confirms NWQDSC as an excellent research field, which can lead to much more important results. Even considering the use of halide perovskites in the device structure, solar cells with vertically aligned nanostructured photoanodes do not compare to the best results of planar versions. On the other hand, in the great majority of the analyzed works, the use of vertically aligned nanostructured electrodes or the use of QDs as an absorbing element have proved successful. This suggests that the complexity of the overall system, which is challenging to accurately model, is preventing the performance of NWQDSCs from reaching not only a very good level but an excellent one. Furthermore, the latest results on solar cells employing cadmium- and lead-free QDs promise a good performance combined with low environmental toxicity; moreover, there is room for further development in more effective NWs-QDs interfaces, with improved surface coverage, charge collection efficiency, and unexplored coupling with perovskite QDs. From the authors’ perspective, future research in the field of NWQDSCs should be based on studying the following factors that influence their light-harvesting efficiency:A detailed analysis of charge mobility in nanowire bodies after surface functionalization with QDs. In the literature, very short nanowire lengths had a negative impact on the overall performance of NWQDSCs, contrary to what was theoretically expected.In-depth research on band gap engineering due to piezo-phototronic effects at the interface between NWs and QDs, with positive effects on solar cell efficiency.The use of photoanodes with embedded quantum dots, as discussed in Section 2.3, has never been attempted because of the expensive physical deposition techniques needed to realize them, but scientific interest remains open.Finally, one overlooked aspect is the combination of nanowires with PQDs, which has proven to be much more stable than the highly performing layered versions.

## Figures and Tables

**Figure 1 materials-16-04297-f001:**
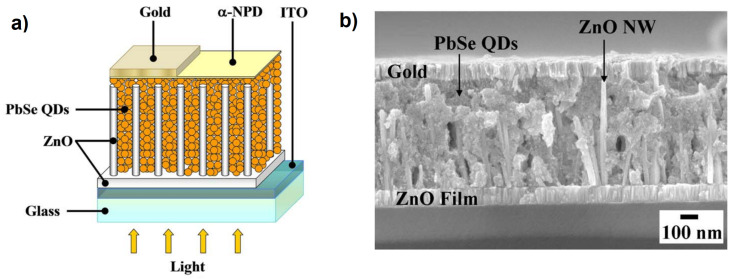
(**a**) Typical device structure of a nanowire–quantum dot solar cell (NWQDSC) as proposed by Leschkies et al. (**b**) Cross-section SEM image of a NWQDSC based on ZnO NWs. Adapted with permission from Ref. [9]. Copyright 2009, American Institute of Physics.

**Figure 2 materials-16-04297-f002:**
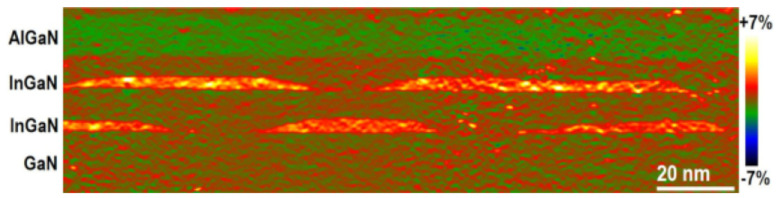
HRTEM image of the out-of-plane strain by the geometric phase analysis of InGaN QDs embedded in the GaN lattice. Reprinted with permission from Ref. [49]. Copyright 2023, MDPI.

**Figure 3 materials-16-04297-f003:**
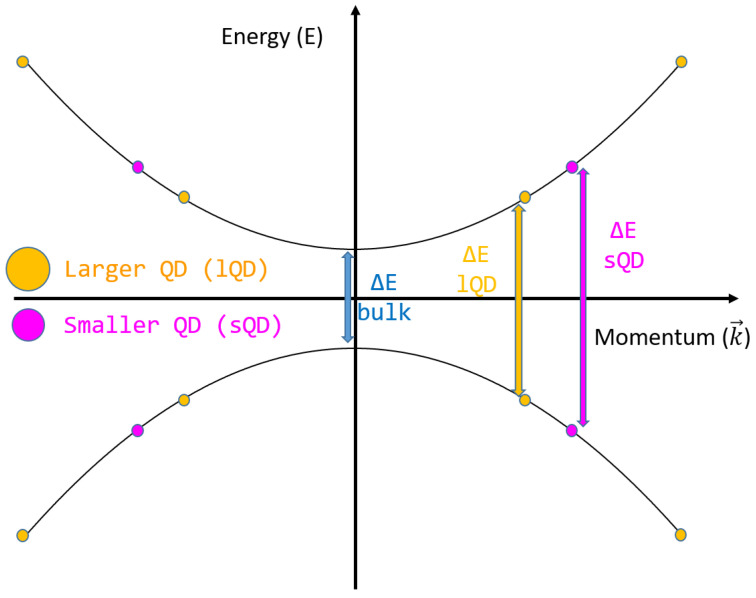
Energy levels and quantum confinement in quantum dots.

**Figure 5 materials-16-04297-f005:**
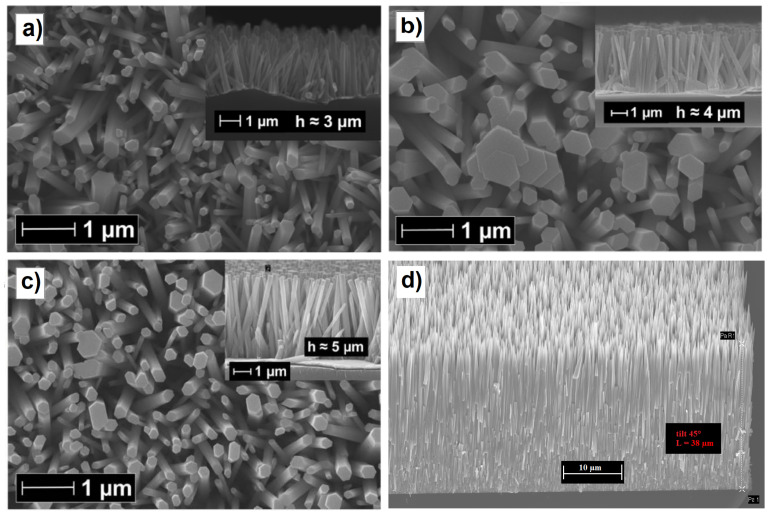
(**a**) Top view SEM image of the ZnO nanorods grown at 90 ${}^{{}^{\circ}}$C with a 2.5 mM equivalent solution of Zn(NO${}_{3}$)${}_{2}$ and HMTA (cross-section in the top-right inset). (**b**) Top view SEM image of the ZnO nanorods grown at 70 ${}^{{}^{\circ}}$C with a 2.5 mM equivalent solution of Zn(NO${}_{3}$)${}_{2}$ and HMTA (cross-section in the top-right inset). (**c**) Top view SEM image of the ZnO nanorods grown by dynamic temperature with a 2.5 mM equivalent solution of Zn(NO${}_{3}$)${}_{2}$ and HMTA (cross-section in the top-right inset). Adapted with permission from Ref. [70]. Copyright 2014, Nature Publishing Group. (**d**) Tilted view SEM image of long ZnO nanorods grown by thermoconvective heating at 90 ${}^{{}^{\circ}}$C with a 30 mM equivalent solution of Zn(NO${}_{3}$)${}_{2}$ and HMTA. Corresponding author, original research item.

**Figure 6 materials-16-04297-f006:**
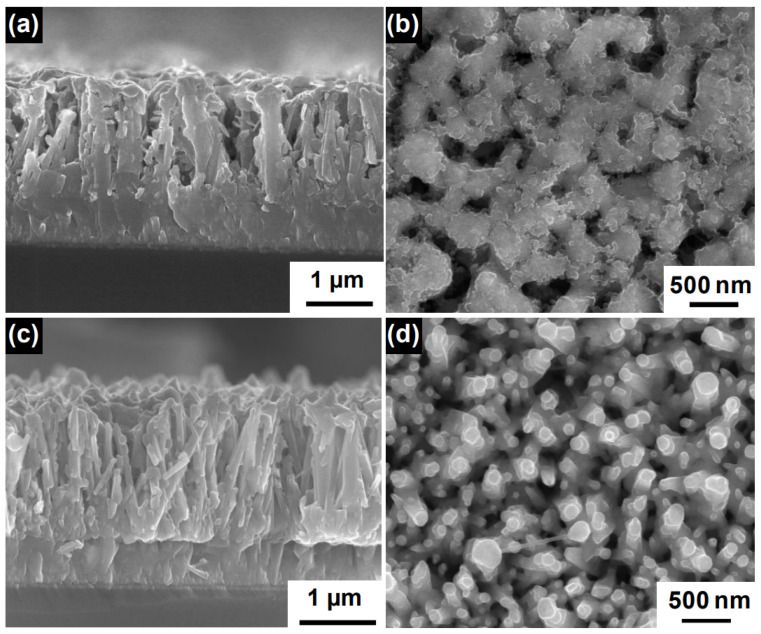
(**a**) Cross-sectional and (**b**) top view images of SnO${}_{2}$/BiVO${}_{4}$ core–shell NWs grown at 500 ${}^{{}^{\circ}}$C, (**c**) cross-sectional and (**d**) top view images of SnO_2_/BiVO_4_ core–shell NWs grown at 600 ${}^{{}^{\circ}}$C. Adapted with permission from Ref. [80]. Copyright 2018, American Chemical Society.

**Figure 7 materials-16-04297-f007:**
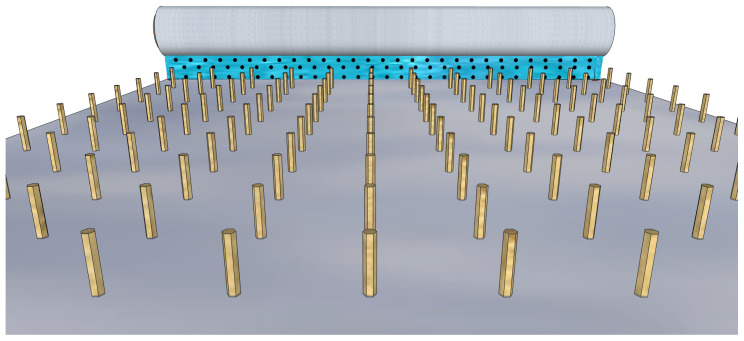
Bar coating a colloidal solution with QDs over a sliding substrate with vertical nanowires.

**Figure 9 materials-16-04297-f009:**
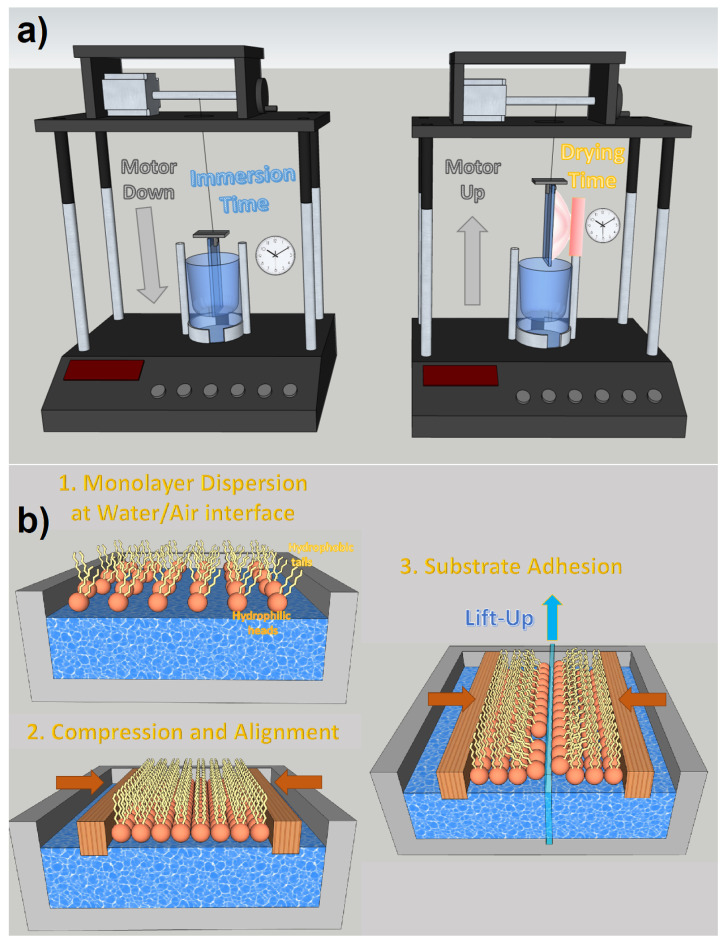
(**a**) Dip coating technique with motorized substrate motions and set-up times for immersion and drying. (**b**) Langmuir–Blodgett deposition steps.

**Figure 10 materials-16-04297-f010:**
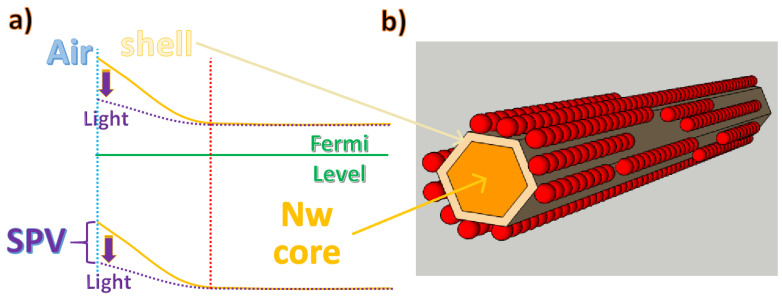
(**a**) The effect of adsorbed acceptor molecules on nanowire electron energy levels. (**b**) Single nanowire superficially functionalized with quantum dots.

**Figure 11 materials-16-04297-f011:**
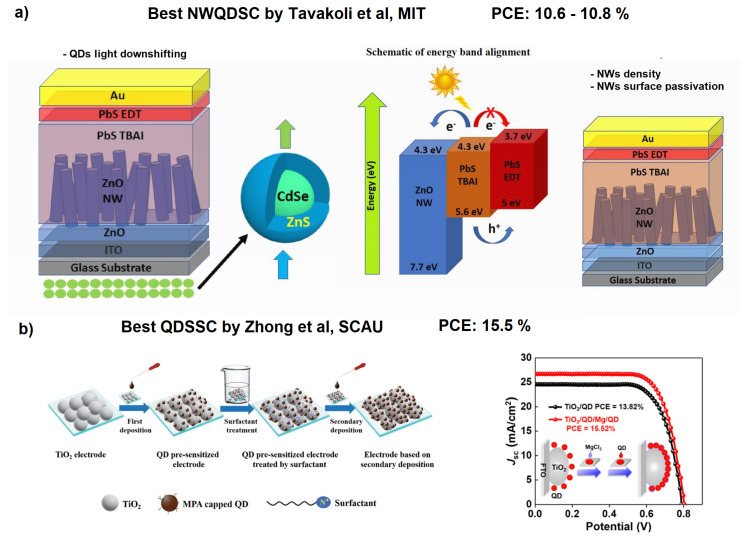
(**a**) Optimized ZnO-based NWQDSCs with frequency down-shifting or density optimization. Reproduced with permission. Copyright 2019, [105] Royal Society of Chemistry. Copyright 2020, [106] Wiley. (**b**) Double ligand-assisted self-assembly approach on TiO${}_{2}$ NPs (drop-casting deposition), allowing increased surface coverage (almost 40% more) and relative solar cell performances. Adapted with permission from Refs. [108,109]. Copyright 2019/2021, American Chemical Society.

**Figure 12 materials-16-04297-f012:**
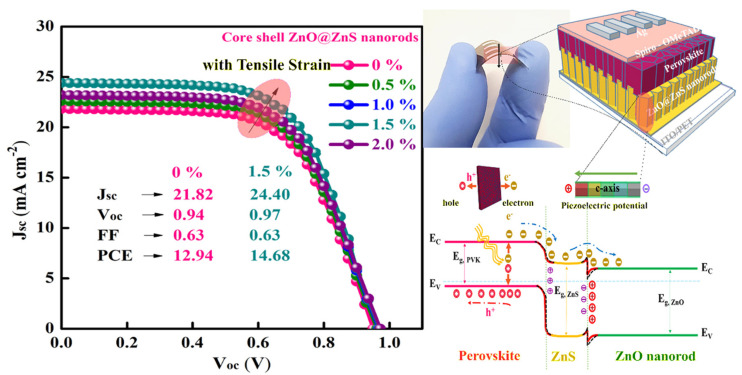
ZnO@ZnS core–shell vertically aligned nanorods on a flexible substrate, allowing PCE improvement from 12.94% to 14.68%, utilizing the piezopotential induced by simple substrate bending. Copyright 2021, [114] Elsevier.

**Table 1 materials-16-04297-t001:** Physical properties of nanostructured semiconducting oxides for solar cells.

Property	ZnO	TiO${}_{2}$	SnO${}_{2}$
Lattice	Wurtzite	Rutile, Atanase	Rutile
Electron mobility * cm${}^{2}$/(Vs)	100–150 [53]	0.001 [54]	>200 [19]
Band gap [55]	3.3 eV	3.2 eV	3.5 eV
Work function [56]	6.4 eV	6.4 eV	6.1 eV

* Not bulk values but measured in nanostructures.

**Table 2 materials-16-04297-t002:** The best solar cells from comprehensive reviews on QDs [17,18] and ZnO NW solar cells [20,96].

Substrate	Light Absorber	Reference	J${}_{SC}$	V${}_{OC}$	PCE
Nanostructures/Passivation	Ligand/Shell	Year	[mA/cm${}^{2}$]	[V]	[%]
Glass/ITO	CuSbS${}_{2}$ QDs	[98]	5.9	0.49	1.61
ZnO NWs		[2016]			
Glass/FTO	CdSe/CdS QDs	[99]	15.6	0.74	5.92
ZnO NWs	TiO${}_{2}$/Ag NPs	[2016]			
Glass/ITO	PbS QDs	[100]	29.4	0.57	9.6
ZnO NWs	TBAI *	[2016]			
Mesoporous	Zn–Cu–In–Se	[101]	24.95	0.62	9
TiO${}_{2}$ NWS		[2017]			
FTO	Mn:In${}_{2}$S${}_{3}$	[102]	26.5	0.644	8
TiO${}_{2}$ nanoparticles	CuInS${}_{2}$	[2018]			
Glass/FTO	PbS QDs	[103]	23.2	0.598	7.49
ZnO NWS/ZnSnO${}_{2}$		[2019]			
Glass/ITO	PbS QDs	[104]	27.9	0.53	9.92
ZnO NWs/4-aminobenzoic acid	TBAI *	[2019]			
Glass/ITO	PbS + CdSe/ZnS QDs	[105]	32.8	0.592	10.6
ZnO NWs	TBAI *	[2019]			
Glass/ITO	PbS QDs	[106]	31.1	0.61	10.8
ZnO NWs/hydrogen plasma	TBAI *	[2020]			
Glass/FTO	Zn–Cu–In–Se	[107]	24.2	0.79	13.5
TiO${}_{2}$ nanoparticles	ZnSe	[2020]			
Glass/FTO	Zn–Cu–In–S–Se	[108]	26.5	0.8	15.3
TiO${}_{2}$ nanoparticles	ZnSe	[2021]			

* TetraButylAmmonium Iodide.

**Table 3 materials-16-04297-t003:** Recent papers on perovskite solar cells based on semiconducting VANWs as ETL.

Substrate,	Reference	J${}_{SC}$	V${}_{OC}$	PCE
Nanostructures/Passivation	Year	[mA/cm${}^{2}$]	[V]	[%]
Glass/ITO, ZnO NWs	[2015], [112]	21.7	0.97	16.2
Glass/AZO, ZnO nanocones/Zn${}_{2}$SnO${}_{4}$	[2019], [113]	23.2	1.07	18.3
Glass/FTO, ZnO nws/ZnSe	[2019], [109]	18.64	0.835	8.2
Flexible PET/ITO, ZnO nws/ZnS	[2021], [114]	24.4	0.97	14.68 *
Glass/FTO, ZnO nws/ZnS	[2021], [115]	23.48	1.1	19.9

* Subjected to mechanical deflections.

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
