# Peer review of "Vertically Aligned Nanowires and Quantum Dots: Promises and Results in Light Energy Harvesting"

_materials, 2023, doi:10.3390/ma16124297_

Round 1

Reviewer 1 Report

In this review, the authors concentrate on the combination of vertically aligned nanowires and quantum dots which exhibit improving photoconversion efficiencies, meanwhile, they summarize kinds of fabrication methods for both materials. This work can be considered for publication after the following issues.

1.     This review remains well conclusion on others’ works but lacks the author’s prospect in this field. It should be better to add this section to the manuscript.

2.     Benefiting from the high photoconversion efficiencies with the combined materials (vertically aligned nanowires and quantum dots), there remain lots of suitable applications, such as photodetectors. The authors remind the photodetectors in the section of keywords, but they fail to elaborate it like the “Nano-Wires Quantum-Dots Solar Cells” which results in the incomplete application. Wang etc. achieve high-performance photodetectors by designing one-dimensional metal-oxides/perovskite heterostructure (refer DOI: 10.1088/1361-6528/ac05e7); Shen etc. reported the fabrication of high-performance ultraviolet photodetectors based on a heterojunction device structure in which ZnO quantum dots were used to decorate Zn2SnO4 nanowires.

3.     In the introduction section, the authors mentioned that “Quantum Dots (QDs) are nano-sized particle with one dimension that is sufficiently low to be comparable with the Bohr radius of the atoms composing their bulk material, i.e. no more than two decades of scale difference”. It should be better to provide a convincing reference.

4.      This manuscript would do well if the authors focus on the mechanistic investigation of the composite of the two materials. Johnny C. Ho etc. introduce a metal-cluster-decoration approach to tailor the electronic transport properties of III-V nanowire field transistors through the modulation of free carriers in the nanowire channel via the deposition of different metal clusters with different work functions (refer DOI: 10.1002/adma.201301362).

There are some grammatical mistakes. The authors should check and revised.

Author Response

Dear Reviewer,

the answer is in the attached PDF.

Our best regards

The authors

Reviewer 2 Report

The paper is a review paper and gives a well summary about a vertically aligned NWs QDs as an energy harvesting, however some comments should be addressed:

- In the abstrct section, a brife summary about the main concuslion should be mentioned.

- The paper title should include a question mark, please remove it. - In Figure 1, the authors should distinguish between the given two sub-figures and give a description of both.

- In section 3 - lines 174-176, the authors need to give the full name of the materials and then they can use the appriviations unstead.

Author Response

(The authors gave the same response as above.)

Reviewer 3 Report

In the manuscript "Vertically aligned nanowires and quantum dots: a perfect match for light energy harvesting" the authors review the use of ZnO, TiO2 and SnO2 in photovoltaic cells functionalized with quantum dots. Unfortunately, I cannot recommend it for publication for the reasons listed below.

1) The authors state in the Conclusions, that "NWQDSCs have been widely studied and reviewed in comparison to similar solar cells without the use of nanostructured anodes (QDSCs or QDSSCs) or without the use of QDs as light absorbing layers (NWSCs)." If NWQDSCs have already been reviewed, what is the novelty of the manuscript under consideration?

2) The manuscript needs thorough language correction to become useful for the reader. There are numerous fragments where the mistakes hamper reading or prevent comprehension of the authors' point. Just a couple of examples:

lines 402-406: "Considering that research on other available energy technologies promising to overwhelm the Shockley–Queisser limit, like thermionic emitters with solar concentration systems is still at embryonal level ( PCE <10% [119]), we can assert NWQDSCs are an excellent research field, starting 14 years ago with a PCE of 2% until achieving efficiencies beyond 10% but still not in line with research on planar solution-processed solar cells."

lines 365-369: "Also considering solar cells based on single-junction synthetic halide perovskites whose overall PCE boosted from 3.8% up to 25.5% from 2009 to today [105], the effect of interpenetrating ZnO NWs does not seem to be so positive if compared, termos of the overall efficiency results, to perovskite solar cells employing planar defect-engineered semiconducting ETL layers."

3) The manuscript needs style correction, as the wordy and diffusive style currently used prevent evaluation of the scientific soundness of this review.

In my view, the manuscript “Vertically Aligned Nanowires and Quantum Dots: a Perfect Match for Light Energy Harvesting” is not suitable for publication for the reasons given below.
The manuscript gives a brief overview on previously reported methods for preparation of quantum dots and vertically aligned nanowires and on fabrication of solar cells based on combination of nanowires with quantum dots. The discussion on the QD/nanowires solar cells takes around 50 lines out of 420 lines of the text (without the reference list). On my opinion, the discussion given there looks more like enumeration of results from other works without proper analysis, hence I do not consider it original enough to warrant the publication. The conclusions section is also written in a rather diffusive style, the authors should make it more succinct to highlight the ideas they derived from the literature review. Besides, the authors should avoid introducing new literature sources in the conclusions. E.g., references 119 and 120 appear only in conclusions. Hence, the conclusions are not consistent with the evidence and arguments presented, rather include additional evidence and arguments themselves.
Unfortunately, I do not think that the manuscript under consideration adds new data or ideas to the previously published materials.  To make the review worth publication, the authors should significantly improve the style and language of the manuscript and present the new ideas arising from the analysis of the literature sources in a clear and succinct style.
Other remarks:
Concerning the references, the reference to the site of Sigma-Aldrich in line 203 is not quite informative. The list of references also needs careful attention as it has a non-uniform style of references, e.g. please compare refs. 1, 3 and 4.
Figures 2, 4, 8 and 10 seem to be original; other figures are reprinted illustrations from other sources, the references are given in the legends. For Fig. 6d no source is mentioned. Is it original? Figure 6 is not mentioned in the text.
The authors state they report SEM images of DSSC in Figure 7. Yet, Figure 7 is claimed to reproduce figure from ref. 77, by other authors. If that is the case, the word “report” is not the best choice. Table 1 has strange notation for units of electron mobility. Also, what does the asterisk at “electron mobility” stand for? 

The manuscript needs thorough language correction to become useful for the reader. There are numerous fragments where the mistakes hamper reading or prevent comprehension of the authors' point. Just a couple of examples:

lines 402-406: "Considering that research on other available energy technologies promising to overwhelm the Shockley–Queisser limit, like thermionic emitters with solar concentration systems is still at embryonal level ( PCE <10% [119]), we can assert NWQDSCs are an excellent research field, starting 14 years ago with a PCE of 2% until achieving efficiencies beyond 10% but still not in line with research on planar solution-processed solar cells."

lines 365-369: "Also considering solar cells based on single-junction synthetic halide perovskites whose overall PCE boosted from 3.8% up to 25.5% from 2009 to today [105], the effect of interpenetrating ZnO NWs does not seem to be so positive if compared, termos of the overall efficiency results, to perovskite solar cells employing planar defect-engineered semiconducting ETL layers."

The manuscript needs style correction as well, as the wordy and diffusive style currently used prevent evaluation of the scientific soundness of this review.

Author Response

(The authors gave the same response as above.)

Round 2

Reviewer 1 Report

The authors have replied all the questions.

Author Response

We thank the reviewer for his helpful comments.

Reviewer 3 Report

The authors have partially addressed my concerns. In particular, they added notion on the novelty of the manuscript (lines 457-460; actually, this fragment is expected in the Introduction section rather than in the Conclusions) and corrected some minor issues. Yet, the style and language issue remains mostly unaddressed. My recommendation is to thoroughly revise the manuscript in terms of language correction.

Dear Authors,

In my view, review papers are mostly intended for the non-experts in the field, like students, newcomers and professionals from adjacent fields. For this reason, the language quality of the review paper is of utmost importance. Please revise the manuscript to facilitate the reading and to present your ideas as clear as possible.

Author Response

The authors thank the reviewer
and the authors will use the maxim
shrewdness in linguistic style
of the next resubmission